# Circulating Cell-Free DNA and RNA Analysis as Liquid Biopsy: Optimal Centrifugation Protocol

**DOI:** 10.3390/cancers11040458

**Published:** 2019-03-30

**Authors:** Laure Sorber, Karen Zwaenepoel, Julie Jacobs, Koen De Winne, Sofie Goethals, Pablo Reclusa, Kaat Van Casteren, Elien Augustus, Filip Lardon, Geert Roeyen, Marc Peeters, Jan Van Meerbeeck, Christian Rolfo, Patrick Pauwels

**Affiliations:** 1Center for Oncological Research Antwerp (CORE), University of Antwerp (UAntwerp), 2610 Wilrijk, Belgium; Karen.Zwaenepoel@uza.be (K.Z.); Julie.Jacobs@uantwerpen.be (J.J.); preclusa.1@alumni.unav.es (P.R.); Kaat.VanCasteren@kuleuven.be (K.V.C.); Elien.Augustus@uantwerpen.be (E.A.); Filip.Lardon@uantwerpen.be (F.L.); Marc.Peeters@uza.be (M.P.); Jan.VanMeerbeeck@uza.be (J.V.M.); Christian.Rolfo@umm.edu (C.R.); Patrick.Pauwels@uza.be (P.P.); 2Laboratory of Pathological Anatomy, Antwerp University Hospital (UZA), 2650 Edegem, Belgium; Koen.DeWinne@uza.be; 3Biobank UZA/UAntwerpen, Antwerp University Hospital (UZA), 2650 Edegem, Belgium; Sofie.Goethals@uza.be; 4Biomedical Quality Assurance Research Unit, KU Leuven, 3000 Leuven, Belgium; 5Hepatobiliary Transplantation and Endocrine Surgery, Antwerp University Hospital (UZA), 2650 Edegem, Belgium; Geert.Roeyen@uza.be; 6Department of Oncology, Multidisciplinary Oncological Center Antwerp, Antwerp University Hospital (UZA), 2650 Edegem, Belgium; 7Department of Pulmonology and Thoracic Oncology, Antwerp University Hospital (UZA), 2650 Edegem, Belgium; 8Thoracic Medical Oncology and Early Clinical Trials, Marlene and Steward Greenebaum Comprehensive Cancer Center, Baltimore, MD 21201, USA

**Keywords:** liquid biopsy, cell-free DNA, cell-free RNA, centrifugation protocol, ddPCR

## Abstract

The combined analysis of circulating cell-free (tumor) DNA (cfDNA/ctDNA) and circulating cell-free (tumor) RNA (cfRNA/ctRNA) shows great promise in determining the molecular profile of cancer patients. Optimization of the workflow is necessary to achieve consistent and reproducible results. In this study, we compared five centrifugation protocols for the optimal yield of both cfDNA/ctDNA and cfRNA/ctRNA. These protocols varied in centrifugation speed, ambient temperature, time, and number of centrifugation steps. Samples from 33 participants were collected in either BD Vacutainer K_2_EDTA (EDTA) tubes or cell-free DNA BCT^®^ (Streck) tubes. cfDNA concentration and fragment size, and cfRNA concentration were quantitated in all samples by digital droplet PCR (ddPCR) and quantitative PCR (qPCR). The *KRAS*-mutated ctDNA and ctRNA fraction was determined via ddPCR. In EDTA tubes, the protocol generating both plasma and platelets was found to produce high quality cfDNA and cfRNA concentrations. Two-step, high-speed centrifugation protocols were associated with high cfDNA but low cfRNA concentrations. High cfRNA concentrations were generated by a one-step, low-speed protocol. However, this coincided with a high amount of genomic DNA (gDNA) contamination. In Streck tubes, two-step, high-speed centrifugation protocols also generated good quality, high cfDNA concentration. However, these tubes are not compatible with cfRNA analysis.

## 1. Introduction

The analysis of circulating cell-free (tumor) DNA (cfDNA/ctDNA) as a liquid biopsy has shown tremendous potential in many aspects of oncology, ranging from minimally invasive molecular profiling and treatment monitoring to the detection of resistance mutations [1]. Several limitations of ctDNA-based liquid biopsies have been addressed over the last few years. Complications include the short half-life of cfDNA [2] as well as wild type (WT) DNA contamination from leukocyte lysis [3]. In order to detect the often-low concentration and highly variable ctDNA fraction, several techniques with high analytical sensitivity and specificity have been developed [4]. Furthermore, blood preservation tubes have been designed, which stabilize white blood cells and inhibit nuclease activity [5]. ctDNA has been established as an excellent source of mutation and methylation analysis [4], however, it might be less appropriate for the detection of gene rearrangements. Even though the group of Russo et al. were successful in detecting gene-fusions in ctDNA [6], extensive deep sequencing or break-point encompassing targeted assays are necessary to detect rearrangements. Circulating cell-free (tumor) RNA (cfRNA/ctRNA) analysis could complement ctDNA in determining the molecular profile. Analysis of cfRNA in circulation or embedded in vesicles or tumor-educated platelets (TEPs) has shown great potential in the detection of several cancer-associated aberrations [7,8,9].

Standardization of the liquid biopsy workflow is necessary for clinical implementation [10]. Several studies have been performed to determine optimal pre-analytical conditions for cfDNA [11,12,13] and, to a lesser extent, for cfRNA analysis [14]. However, studies which address the generation of both cfDNA and cfRNA are lacking. In this study, we compared five centrifugation protocols with varying centrifugation speed, temperature, time, and number of centrifugation steps for the optimal cfDNA and cfRNA yield. These protocols were evaluated in two of the most commonly used blood collection tubes, namely BD Vacutainer K_2_EDTA tubes (EDTA tubes; BD, Erembodegem, Belgium) and cell-free DNA BCT^®^ tubes (Streck tubes; Streck, Biomedical Diagnostics, Antwerp, Belgium). To this purpose, both cfDNA and cfRNA were isolated from healthy volunteers and patients with *KRAS*-mutated metastatic cancer. CfDNA concentration and fragment size, and cfRNA concentration were quantitated in all samples. The fraction of *KRAS*-mutated ctDNA and ctRNA was also determined.

## 2. Results

In this study, five centrifugation protocols were compared for the optimal yield of both cfDNA/ctDNA and cfRNA/ctRNA in either EDTA or Streck tubes (Table 1).

### 2.1. Total cfDNA Concentration and Characteristics

The cfDNA yield, as well as genomic DNA (gDNA) contamination, were examined per centrifugation protocol. Significant differences in total cfDNA concentration were observed by digital droplet PCR (ddPCR) (*p* < 0.001 and *p* = 0.004) and quantitative PCR (qPCR) analysis (*p* = 0.001 and *p* = 0.017) in both EDTA and Streck tubes, respectively. Centrifugation protocols were also found to influence DNA integrity in EDTA (*p* < 0.001) and Streck tubes (*p* < 0.001), as determined by fragment size analysis. The individual differences between these protocols was further investigated (Figure 1; individual *p*-values are provided in Appendix A).

#### 2.1.1. cfDNA Yield

DdPCR analysis revealed that the highest cfDNA concentrations in EDTA tubes were generated by the basic (CP_Basic_; one-step, low-speed) centrifugation protocol and the pellet of the adapted basic protocol (CP_AdBasic_P_). No significant differences were observed between the other centrifugation protocols (Figure 1A). Similar results were observed by qPCR analysis (Figure 1B).

In Streck tubes, ddPCR analysis revealed that the basic centrifugation protocol (CP_Basic_) alone was responsible for the highest cfDNA concentration. The pellet of the adapted basic protocol (CP_AdBasic_P_) generated a very variable cfDNA yield and did not differ from the other centrifugation protocols (Figure 1A). More differences were observed with qPCR (Figure 1B). cfDNA concentration seemed to be similar between the basic protocol (CP_Basic_), the Streck (CP_Streck_), and adapted CEN (CP_AdCEN_) protocols. However, while the basic protocol had a significantly higher cfDNA yield than the platelet-generating protocol (CP_plat_), the plasma (CP_AdBasic_), and pellet (CP_AdBasic_P_) of the adapted basic protocol, only the latter (CP_AdBasic_P_) was found to be lower than that of the Streck (CP_Streck_) and adapted CEN (CP_AdCEN_) protocols.

#### 2.1.2. cfDNA Integrity

Next, we investigated DNA integrity (Figure 1C). In EDTA tubes, the pellet of the adapted basic protocol (CP_AdBasic_P_) generated the highest concentration of long fragments (gDNA contamination), followed by the basic protocol (CP_Basic_) itself. The plasma of the adapted basic protocol generated significantly fewer long fragments, but still retained higher concentrations than the other centrifugation protocols. Interestingly, no differences in DNA integrity were detected between the platelet generating (CP_Plat_) and (adapted) CEN (CP_(Ad)CEN_) protocols, despite their differences in centrifugation speed and time. 

Similar results were detected in Streck tubes. The highest concentration of long fragments was detected in the pellet of the adapted basic protocol (CP_AdBasic_P_), followed by the basic protocol (CP_Basic_). However, there was no significant difference in DNA integrity between the plasma of the adapted basic (CP_AdBasic_) and the platelet-generating (CP_Plat_) protocol. As seen in EDTA tubes, the two-step, high-speed (adapted CEN (CP_AdCEN_), Streck (CP_Streck_)) protocols, as well as the two-step, low-speed, platelet-generating protocol (CP_Plat_), were found to generate good-quality cfDNA (low amounts of long fragments).

Even though comparing the performance of EDTA and Streck tubes was not part of the aim of this study, a significant difference in cfDNA concentration and DNA integrity per centrifugation protocol was detected between these tubes, with the exception of the tube specific protocols (CP_CEN_ and CP_Streck_). The pellet of the adapted basic protocol (CP_AdBasic_P_) generated significantly higher cfDNA concentrations in EDTA tubes according to ddPCR (*p* = 0.015) and qPCR (*p* < 0.001) analysis. Similar distributions were observed in the basic protocol (CP_Basic_), although, only by qPCR analysis (*p* = 0.013). In contrast, DNA integrity was found to be significantly different in all centrifugation protocols, except for the pellet of the adapted basic protocol (CP_AdBasic_P_) (Figure 2; individual *p* values are provided in Appendix A). Finally, a higher percentage of long fragments was detected in EDTA tubes compared to Streck tubes. 

### 2.2. KRAS-Mutated ctDNA

To investigate the effect of the centrifugation protocol on the ctDNA fraction, we quantified the *KRAS-*mutated tDNA and ctDNA in spiked samples from healthy volunteers and *KRAS*-mutated metastatic cancer patients, respectively. In a first phase, spiked *KRAS*-mutated tDNA was evaluated by ddPCR analysis in samples from 8 participants collected in EDTA (4) and Streck (4) tubes. Next, the same number of samples were collected from *KRAS*-mutated metastatic cancer patients. As the results from spiked and cancer patient samples differed significantly, it was opted to proceed with the data from cancer patient samples only (Figure 3). All samples were investigated for the presence of *KRAS* mutations. In 7 out of 8 patients, *KRAS*-mutated ctDNA could not be detected in the pellet of the adapted basic protocol (CP_AdBasic_P_). The exception was patient 4, who had a very high mutated *KRAS* allele frequency (AF; ranging from 20.6 to 51.9% in the other protocols). No significant differences in *KRAS* ctDNA concentration or AF could be observed between the other centrifugation protocols. However, the sample generated by the basic protocol (CP_Basic_) of patient 2 was defined as *KRAS* WT, while *KRAS*-mutated ctDNA was detected in the plasma generated by the other centrifugation protocols. Despite the number of patients, it seems obvious that both the basic protocol (CP_Basic_) and the pellet of the adapted basic protocol (CP_AdBasic_P_) are not suitable for ctDNA analysis.

### 2.3. Total cfRNA Concentration

Next, the total cfRNA yield was determined using ddPCR (Figure 4). In EDTA tubes, significant differences in cfRNA concentration were observed between the different centrifugation protocols (*p* < 0.0001; individual *p*-values are provided in Appendix A). As expected, platelets (CP_Plat_P_) provided the highest cfRNA yield, while the platelet-rich plasma (PRP) basic protocol (CP_Basic_) still generated more cfRNA than the other centrifugation protocols. No differences could be observed between the pellet (CP_AdBasic_P_) and the platelet-poor plasma (PPP) of the adapted protocol (CP_AdBasic_) and the platelet-generating protocol (CP_Plat_). These protocols still generated higher cfRNA concentrations compared to the adapted and original CEN protocols (CP_(Ad)CEN_), which were also found to be similar in cfRNA yield. In Streck tubes, no cfRNA or only very low concentrations could be detected across all centrifugation protocols.

### 2.4. KRAS-Mutated ctRNA

To further investigate the effect of the centrifugation protocol on the ctRNA fraction, the samples were also screened for the presence of *KRAS* mutations. In EDTA tubes, *KRAS*-mutated ctRNA could only be detected in the platelets (CP_Plat_P_) (Figure 5) in 3 out of 4 *KRAS*-mutated metastatic cancer patients (as depicted in Figure 3). In Streck tubes, no *KRAS*-mutated ctRNA could be detected in any centrifugation protocol, as only extremely low cfRNA concentrations, or no cfRNA at all, were generated.

## 3. Discussion

The purpose of this study was to determine the optimal centrifugation protocol for the generation of high-quality cfDNA and cfRNA in two commonly used blood collection tubes; EDTA and Streck tubes. Despite the relatively small sample size of this study, significant differences in cfDNA and cfRNA yield and quality could be observed. In EDTA tubes, the plasma and platelets from the platelet-generating protocol were found to produce both good quality cfDNA and high cfRNA concentration. cfDNA analysis in Streck tubes generated similar results in all centrifugation protocols, however, at best, very low levels of cfRNA could be generated.

In EDTA tubes, the cfDNA characteristics, namely cfDNA concentration and contamination with long DNA fragments, from the platelet-generating protocol were found to be similar to those of the adapted and original CEN protocols. *KRAS*-mutated ctDNA concentration and AF also seemed similar. While the platelet-generating protocol consists of two slow-speed centrifugation steps, the (adapted) CEN protocols have a two-step, high-speed procedure. Most studies recommend a double centrifugation protocol, consisting of an initial slow centrifugation to separate plasma from buffy coat, followed by a fast centrifugation to clear any remaining cellular material and generate high quality cfDNA [12,13,15]. Although there is no consensus with regards to actual centrifugation speed [16,17], both centrifugation steps described in these studies, together with the adapted and original CEN protocols, are much faster in comparison to the platelet-generating protocol. We speculate that the longer centrifugation time (20 min versus 10 min) might be responsible for the low contamination with long DNA fragments, despite the slow centrifugation speed. In contrast, the plasma of the adapted basic protocol, generated by a first slow-speed centrifugation step, followed by a 1-min high-speed centrifugation post-thaw, was found to contain significantly more long DNA fragments. Hence, the interplay between centrifugation speed and time probably has a stronger influence on cfDNA generation than these other factors alone. We did not detect an influence of temperature, as both cfDNA characteristics and ctDNA detection were similar between the adapted and original CEN protocols. Furthermore, no differences between these four centrifugation protocols could be observed in mutated ctDNA concentration and AF. It is important to note that, despite our limited number of patients with detectable *KRAS*-mutated ctDNA, the basic protocol and the pellet of the adapted basic protocol were found to be less suitable for cfDNA analysis. The pellet of the adapted basic protocol seems to consist almost exclusively of long DNA fragments. This is to be expected, as previous research highlights that the second centrifugation step removes any remaining cellular material [12]. Hence, the resulting plasma will mostly contain the cfDNA fraction. Our data support this, as *KRAS*-mutated ctDNA was only detected in the pellet of a patient with a very high ctDNA concentration. The mutated *KRAS* ctDNA AF in the other centrifugation protocols ranged between 20.6% and 50.9%. This is highly unusual, as ctDNA is most often present in very low (<1%) percentages [2]. The plasma of the basic protocol contains both the remaining cellular material as well the cfDNA fraction, as indicated by the high percentage of long fragments. In this manner, the ctDNA is diluted by WT DNA, resulting in the classification of one patient sample as WT despite the use of the highly sensitive ddPCR technique [18]. Furthermore, there are several disorders and physiological processes, such as autoimmune disorders, exhaustive exercise, or trauma, that are known to contribute to the total cfDNA concentration [19]. This might explain the low AF detected in all centrifugation protocols from patient 7 (Figure 3). In these cases, further dilution of the mutated ctDNA with WT gDNA would be detrimental to analytical success.

In Streck tubes, cfDNA analysis revealed similar results as in EDTA tubes. Interestingly, no difference in the generated plasma volume was observed in EDTA or Streck tubes. This was unexpected, especially in the case of the Streck tubes. In a multicenter study, we observed a negative influence on plasma volume upon longer storage of blood samples in Streck tubes. Increased centrifugation speed was found to negate this effect [20,21], which is recommended by Streck. As plasma volume is directly associated with cfDNA concentration [22], it is doubtful whether protocols based on low centrifugation speed, such as the platelet-generating and (adapted) basic protocol, will generate sufficient material for the detection of low abundant aberrations. Even though there were no significant differences in cfDNA characteristics detected between the platelet-generating protocol and the Streck and adapted CEN protocol, it seems that there was still a slight shift towards more long fragments in the former. This might explain the fact that no significant differences were observed between this protocol and the plasma of the adapted basic protocol. The Streck and adapted CEN protocols seem to be the most appropriate centrifugation protocols for cfDNA analysis in Streck tubes. Both are two-step, high-speed protocols, with a first centrifugation step of <2000 *g* for 10 min. However, the second step differs significantly (6000 vs. 16,000 g). This corresponds to previously reported data in which no difference in cfDNA concentration was observed between a second centrifugation step of 3000 *g* or >10,000 *g* [17]. Hence, there is no need to have access to a full spectrum of centrifuges with higher centrifugation speeds. Interestingly, significant differences in percentage of long fragments were observed between centrifugation protocols of EDTA vs. Streck tubes, with the exception of the pellet of the adapted basic protocol. Van Dessel et al. reported a background level of leukocyte lysis, which is responsible for low levels of gDNA [11]. Hence, leukocyte lysis might be diminished by the cell stabilizing effect of Streck tubes [21].

In EDTA tubes, the highest cfRNA concentration, as well as *KRAS*-mutated ctRNA, were detected in the platelet fraction of the platelet-generating protocol. Our data corresponds to previous research in which platelets were highlighted to be an excellent source for tumor RNA (tRNA) [7,8]. Nilsson et al. also reported a higher sensitivity for the detection of tumor-associated aberrations in platelet RNA than in the plasma of the platelet-generating protocol. Even though no *KRAS*-mutated ctRNA could be retrieved from the other centrifugation protocols, the basic protocol might be the best alternative in the absence of platelets. The proposed workflow for PPP consists of two centrifugation steps prior to freezing; namely to remove the bulk of circulating cells and the residual platelets [23]. Hence, the basic protocol most likely still contains some platelets due to its single slow-speed centrifugation procedure. Furthermore, a single freeze/thaw cycle of plasma samples containing (residual) platelets has been demonstrated to release significant quantities of miRNA and platelet microparticles [24]. In contrast, the adapted and original CEN protocols both generate PPP, resulting in the lowest cfRNA concentrations of all protocols. Similar cfRNA yields were detected between the plasma of the platelet-generating protocol, and the pellet and plasma of the adapted basic protocol, despite differences in centrifugation speed and time. These findings also highlight the importance of both factors on cfDNA and cfRNA generation. It is important to note that in this study, the entire platelet fraction was used for cfRNA isolation. This fraction was generated from the entire blood sample (10 mL). In the case of plasma samples, only 200 µL of plasma could be used, which corresponds to approximately 500 µL of blood. The pellet of the adapted basic protocol was re-dissolved in 1 mL of nuclease-free water, of which 200 µL was also is used. Hence, this represents approximately 2 mL of blood. By increasing the input volume, higher cfRNA concentrations might be obtained [25]. Recently, RNA isolation kits with increased input volumes have been developed and are currently being tested, among other pre-analytical variables, in a Belgian extracellular RNA quality control (exRNAQC) study [26]. These findings will aid in the standardization of the cfDNA- and cfRNA-based liquid biopsy workflow. Another issue that needs to be addressed is the low *KRAS*-mutated ctRNA concentration and AF compared to ctDNA. This might be the result of the reactivity of isolated platelets, or contamination with residual cells and microparticles [27]. Red blood cells have been shown to contain a plethora of RNAs [28], which might interfere with these results. Furthermore, centrifugation alone might not be sufficient to generate a pure platelet fraction. The group of Rikkert et al. demonstrated that the pellet of the platelet generating protocol not only contains platelets (71%), but also large (tumor-derived) extracellular vesicles (EVs; 22%), circulating tumor cells (CTCs), smaller (tumor-derived) EVs, and cfDNA (<3%) [29]. Even though we were able to detect *KRAS*-mutated ctRNA in platelets in 3 out of 4 patients with detectable *KRAS*-mutated ctDNA, further research is necessary to increase the quality of cfRNA from platelets and plasma.

However, the advantage of the platelet-generating protocol is the generation of sufficient cfRNA concentration and high quality cfDNA from 10 mL of blood. In this manner, the amount of molecular information would be maximized with limited material from the patient. The true potential of cfRNA analysis lies in the detection of gene fusions and amplifications, for which cfDNA is a less appropriate matrix [7]. An example of the potential of this approach is the real-time follow up of *Anaplastic Lymphoma Kinase* (*ALK*) rearranged non-small cell lung cancer (NSCLC). *ALK*-rearranged NSCLC is a validated molecular target of ALK tyrosine kinase inhibitors (TKIs) [30]. Sadly, ALK TKI resistance is an issue, with acquired *ALK* mutations responsible for a large subset of resistant cases [31]. In these cases, it is an advantage to have the cfRNA as well as the cfDNA. The high quality cfDNA can be screened for any resistance mutations, while translocation can be detected in the platelet cfRNA by targeted assays [7] or via sequencing [32].

In contrast, little to no cfRNA was generated in Streck tubes, irrespective of centrifugation protocol. It is likely that the cell-free DNA BCT^®^ (Streck) tubes do not sufficiently stabilize cfRNA. This could be expected, as cell-free RNA BCT^®^ tubes have already been developed [33]. The low yield might also indicate that the preservative of the Streck tubes is not compatible with the cfRNA isolation kit we used. Presently, there are studies ongoing in which the efficiency of several cfRNA tubes and isolation techniques are being evaluated [26,34]. A blood collection tube which preserves both cfDNA and cfRNA would maximize the detection of several cancer-associated aberrations based on one blood sample.

## 4. Materials and Methods

### 4.1. Study Design

In total, 15 patients with *KRAS-*mutated advanced cancer and 18 healthy volunteers were included. Each participant contributed five 10 mL blood samples, which were collected in either EDTA (*n* = 95) (BD, Erembodegem, Belgium) or Streck (*n* = 70) (Streck, Biomedical Diagnostics, Antwerp, Belgium) tubes. In the former, samples were processed within 2 h after blood collection, whereas Streck tubes were left at room temperature for 24 h before plasma generation. The study was conducted according to the Declaration of Helsinki, and ethical committee approval (B300201422715) was obtained from the ethical committee of Antwerp University Hospital (UZA). All participants provided written informed consent.

This study consisted of three phases (Table 2). Healthy volunteers were included in phase one. In phase two, the blood samples from healthy volunteers were spiked with *KRAS*-mutated tumor DNA in order to simulate ctDNA. DNA was isolated from *KRAS*-mutated NSCLC formalin-fixed paraffin embedded (FFPE) tissue samples, because in contrast to high-quality cell line-derived DNA, the fragmented and partly degraded properties of FFPE-derived DNA are a more suitable ctDNA alternative. The resulting spike mix consisted of 900 mutated DNA copies per µL, of which 1 µL was injected into the blood samples immediately after blood collection. The third phase consisted of known *KRAS*-mutated NSCLC, colorectal cancer (CRC), and pancreatic ductal adenocarcinoma (PDAC) patients to validate our findings. Total cfDNA and cfRNA concentration and cfDNA fragment analysis were investigated in all phases. 

Plasma was generated from matched blood samples with different centrifugation protocols varying in centrifugation speed, temperature, time, and number of centrifugation steps (Table 1). This study was performed in collaboration with the Biobank UZA/UAntwerpen. Therefore, the standard (CP_Basic_) protocol from the Biobank UZA/UAntwerpen was included. As most studies recommend two centrifugation steps [12,13,15], an adapted version of this protocol was also added, of which both the plasma (CP_AdBasic_) and resulting pellet (CP_AdBasic_P_) were analyzed. We also selected the “Isolated circulating cell-free DNA from plasma protocol, CEN/TS 16835-3” as advised by the European committee for standardization (CEN; CP_CEN_). This protocol requires for blood samples to be put on ice immediately after blood collection and centrifugation at 4 °C. The adapted protocol (CP_AdCEN_) can be completely performed at room temperature. As Streck tubes are not compatible with low temperatures, we included the manufacturer’s instructed protocol for Streck tubes (CPStreck) [21]. Lastly, we selected a centrifugation protocol which produces both plasma (CP_Plat_) and platelets (CP_Plat_P_) [7]. Plasma samples were stored at −80 °C by the Biobank UZA/UAntwerpen (Antwerp, Belgium; ID: BE71030031000; Belgian Virtual Tumourbank funded by the National Cancer Plan, BBMRI-ERIC; No. Access: 1; Last: 13 June 2018) [35].

### 4.2. cfDNA Isolation and Analysis

cfDNA isolation was performed using the Maxwell RSC ccfDNA large volume plasma kit (Promega, Leiden, The Netherlands) according to the manufacturer’s protocol. Detailed procedures are described elsewhere [20]. Plasma samples were processed immediately after thawing, with the exception of the adapted basic protocol. Prior to processing, the plasma was centrifuged at maximum speed for one minute. The plasma was separated from the resulting pellet, which was hereafter dissolved in 1 mL nuclease-free water. For cfRNA, 0.5 mL of all plasma samples and the dissolved pellet were reserved.

Samples were screened with the ddPCR *KRAS* Screening Multiplex Kit (Bio-Rad, Temse, Belgium) to determine total cfDNA concentration and *KRAS* mutational status. In total, 10 µL of template was used per sample. Detailed procedures can be found elsewhere [36]. The concentration of these samples was not determined prior to ddPCR analysis. Total cfDNA was also quantified by a qPCR assay consisting of an 82-base pair (bp) *Long Interspersed Nuclear Elements* (*LINE-1*) sequence [37]. The samples were brought to the same concentration for cfDNA sample quality assessment using a 425:82 bp ratio. The 425 bp amplicon was designed to encompass the 82 bp amplicon; representing mainly genomic DNA released from blood cells and cfDNA, respectively. Hence, a high ratio is indicative of gDNA contamination [38]. A long fragment contamination curve consisting of cell line and digested FFPE tissue material was used to determine the percentage of long fragment contamination. All samples were run in duplicate. qPCR mixtures were assembled by 4 µL LightScanner Mastermix (Bioké, Leiden, The Netherlands), 1 µL of 1 µM primer sets, and 5 µL (diluted) cfDNA. Cycle conditions were 95 °C for 2.5 min, then 95 °C for 30 s, 58 °C for 30 s, 72 °C for 1 min cycled 45 times, and 72 °C for 3 min. The LightCycler 480 SW (Roche, Vilvoorde, Belgium) was used to calculate the quantitation cycle (Cq).

### 4.3. cfRNA Isolation and Analysis

cfRNA was isolated from 200 µL of plasma or directly from platelets and converted to complement DNA (cDNA) using the miRNeasy Serum/Plasma Kit and miScript II RT Kit, respectively, as per manufacturer’s instructions (Qiagen, Antwerp, Belgium). Samples were analyzed using the ddPCR *KRAS* Screening Multiplex kit in the same manner as the cfDNA samples, with the exception of the platelet cDNA. Only 1 µL of template was used instead of 10 µL.

### 4.4. Statistical Analysis

The Kruskal–Wallis test was performed to ascertain whether there was a significant difference between the centrifugation protocols and blood collection tubes for total cfDNA and cfRNA yield, percentage of long DNA fragments, and *KRAS*-mutated ctDNA fraction. The Mann–Whitney *U* test was used to determine the effect of the individual centrifugation protocols. Statistical analysis was performed using IBM SPSS Statistics version 24.0 (IBM, Brussels, Belgium). *p*-values of <0.05 were considered statistically significant. Figures were made with Graphpad Prism7 (Graphpad Software Inc., La Jolla, CA, USA).

## 5. Conclusions

To conclude, the platelet-generating protocol provides high quality cfDNA and cfRNA concentrations in EDTA tubes. However, the selected centrifugation protocol will be dependent on the downstream analysis. Therefore, we recommend this protocol when looking at cfRNA or multiple aberration types (e.g., translocations and acquired resistance mutations). In the case of cfDNA screening alone, the adapted CEN protocol (two-step, high-speed) would be more appropriate, as it is a shorter and less laborious procedure. In terms of biobanking, where blood samples can be collected without a definite downstream analysis, the basic protocol (one-step, low-speed) might offer the most possibilities. In this case, the generated platelet-rich plasma will permit cfRNA isolation, while a second high-centrifugation step will clear away most of the remaining cellular material. Cell-free DNA BCT^®^ (Streck) tubes should only be used for cfDNA analysis. Both the Streck and adapted CEN protocols generate high quality cfDNA. Further research might reveal blood preservation tubes which generate both good quality cfDNA and cfRNA from one blood sample. 

## Figures and Tables

**Figure 1 cancers-11-00458-f001:**
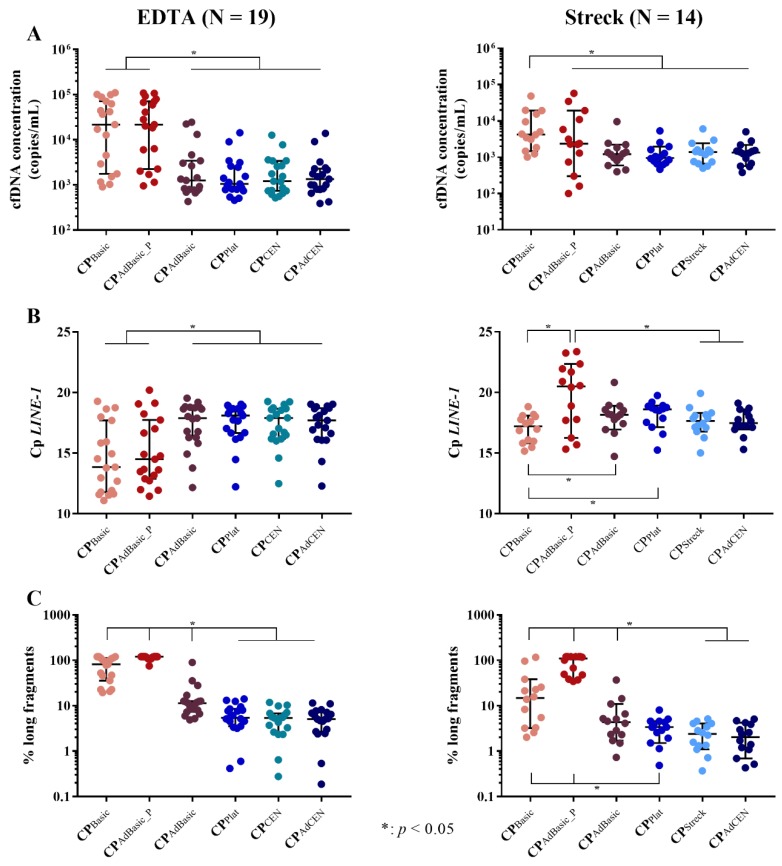
Comparison of total cfDNA yield per centrifugation protocol in EDTA and Streck tubes. In all samples, (**A**) digital droplet PCR (ddPCR) (copies/mL) and (**B**) *LINE-1* 82 bp quantitative PCR (qPCR) were performed to determine total cfDNA concentration, as well as (**C**) cfDNA integrity analysis by qPCR to determine the percentage of long (> 400 bp) fragments; N: number of participants.

**Figure 2 cancers-11-00458-f002:**
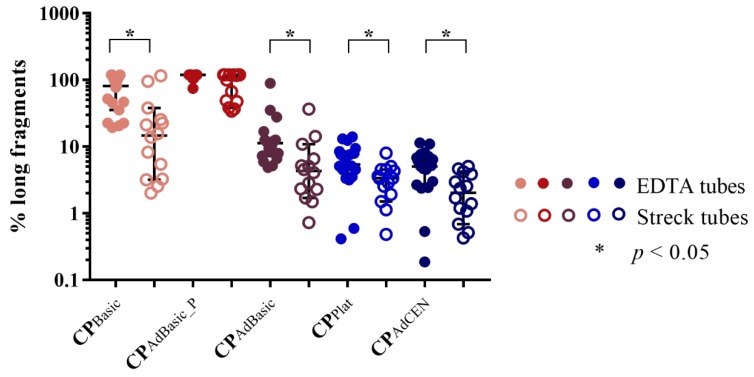
cfDNA integrity in EDTA (N = 19) vs. Streck (N = 14) tubes; N: number of participants.

**Figure 3 cancers-11-00458-f003:**
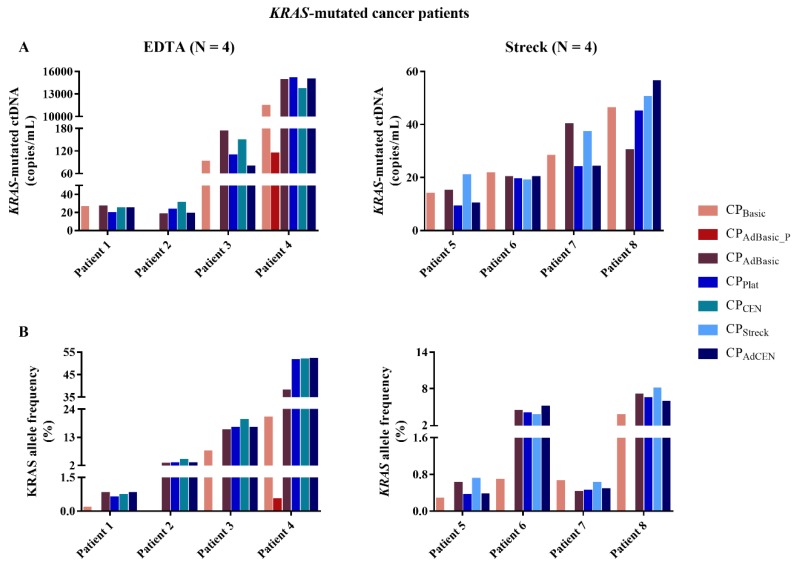
ctDNA analysis of *KRAS*-mutated metastatic cancer patients (8) per centrifugation protocol (Table 1) in EDTA (left) and Streck (right) tubes. (**A**) Concentration of *KRAS*-mutated ctDNA (copies/mL); (**B**) *KRAS*-mutated ctDNA allele frequency (%); N: number of participants.

**Figure 4 cancers-11-00458-f004:**
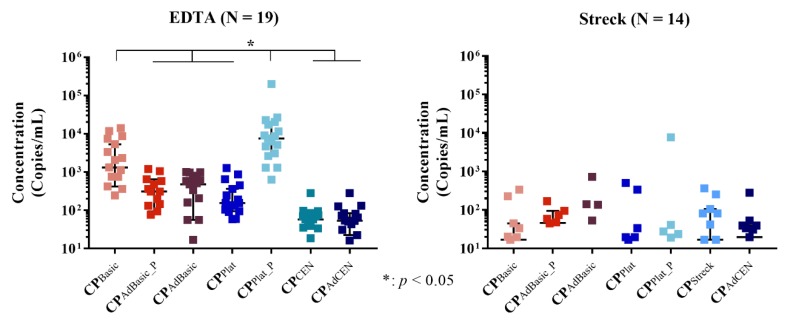
Comparison of cfRNA yield per centrifugation protocol in EDTA and Streck tubes; N: number of participants.

**Figure 5 cancers-11-00458-f005:**
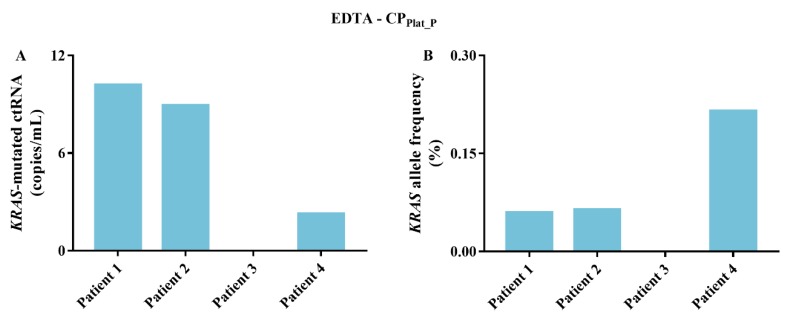
ctRNA analysis of *KRAS*-mutated metastatic cancer patients (4). (**A**) Concentration of *KRAS*-mutated ctRNA (copies/mL); (**B**) *KRAS*-mutated ctRNA allele frequency (%).

**Table 1 cancers-11-00458-t001:** Centrifugation protocols for cfDNA/cfRNA analysis.

ID	Protocol	Specifications	Details	Temperature	Matrix
CP_Basic_	Basic (Biobank UZA/UAntwerpen)	1. 10’ – 400 *g*		RT	Plasma
CP_AdBasic_P_	Basic (adapted)	1. 10’ – 400 *g* 2. 1’ – max speed	2nd centrifugation step after storage at −80 °C	RT	Pellet
CP_AdBasic_					Plasma
CP_Plat_	Platelet	1. 20’ – 120 *g* 2. 20’ – 360 *g* 3. 5’ – 360 *g*	Platelets are washed with PBS in 3rd centrifugation step	RT	Plasma
CP_Plat_P_					Platelets
CP_Streck_	Streck	1. 10’ – 1600 *g* 2. 10’ – 6000 *g*		RT	Plasma
CP_CEN_	CEN	1. 10’ – 1900 *g* 2. 10’ – 16,000 *g*	Blood samples on ice immediately after blood collection	4 °C	Plasma
CP_AdCEN_	CEN (adapted)	1. 10’ – 1900 *g* 2. 10’ – 16,000 *g*		RT	Plasma

A smooth braking profile was used in all centrifugation protocols to prevent disruption of the buffy coat layer. Plasma (and platelets) from each centrifugation protocol were snap frozen and stored at −80 °C. CEN: European Committee for Standardization; cfDNA: circulating cell-free DNA; cfRNA: circulating cell-free RNA; CP: centrifugation protocol; Platelets: only RNA isolation; RT: room temperature; 1: first; 2: second, and; 3: third centrifugation step, respectively.

**Table 2 cancers-11-00458-t002:** Study cohort.

Phase	Participants	EDTA (*N* = 19)		Streck (*N* = 14)	
		cfDNA Samples	cfRNA Samples	cfDNA Samples	cfRNA Samples
1	Healthy volunteers	n° = 48	n° = 55	n° = 12	n° = 7
cfDNA: total cfDNA concentration quantification and fragment analysis; cfRNA: total cfRNA concentration quantification.					
2	Healthy volunteers—spiked with *KRAS-*mutated DNA	n° = 24	n° = 28	n° = 24	n° = 28
cfDNA: total cfDNA concentration quantification and fragment analysis; spiked tumor DNA: tDNA concentration and allele frequency quantification; cfRNA: total cfRNA concentration quantification.					
3	*KRAS*-mutated metastatic cancer patients	n° = 42	n° = 47	n° = 48	n° = 53
cfDNA: total cfDNA concentration quantification and fragment analysis; ctDNA: ctDNA concentration and allele frequency quantificationcfRNA: total cfRNA concentration quantificationctRNA: ctRNA concentration and allele frequency quantification.					

cfDNA: circulating cell-free DNA; cfRNA: circulating cell-free RNA; ctDNA: circulating cell-free tumor DNA; ctRNA: circulating cell-free tumor RNA; N: number of subjects; n°: number of samples; tDNA: tumor DNA.

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
