# Peer review of "Circulating Cell-Free DNA and RNA Analysis as Liquid Biopsy: Optimal Centrifugation Protocol"

_cancers, 2019, doi:10.3390/cancers11040458_

Round 1
Reviewer 1 Report
The article covers an interesting area of precision medicine, addressing technical issues in liquid biopsy pre-analytical and analytical optimization.
My only concern regards the paragraph KRAS mutated ctDNA, which is really difficult to follow. It would be auspicable to specify in a more linear way how the presence of mutant RAS ctDNA was
detected, how it was quantified (how many ng of DNA were used?) and how authors assesses the quality of ctDNA. Why authors did not use mutant RAS spiked cells as positive control?
Please improve english
Author Response
Dear Editor
Dear Reviewer
Please find enclosed the amended version of our manuscript 'Circulating Cell-Free DNA and RNA Analysis as Liquid Biopsy: Optimal Centrifugation Protocol'. We very much appreciate the constructive comments of the reviewer.
We are pleased to submit the improved manuscript within the requested deadline. We have adapted our manuscript according to the reviewer's recommendations, as described by our point-by-point reply below:
1. My only concern regards the paragraph KRAS mutated ctDNA, which is really difficult to follow. It would be auspicable to specify in a more linear way how the presence of mutant RAS ctDNA was detected, how it was quantified (how many ng of DNA were used?) and how authors assesses the quality of ctDNA. Why authors did not use mutant RAS spiked cells as positive control?
As per the reviewers' suggestion, we have rephrased the paragraph "2.2. KRAS mutated ctDNA" to describe the KRAS ctDNA analysis in a more straightforward way [Lines 139 – 147 of the revised manuscript]. Concerning the use of mutant cells as positive controls, we have opted not to proceed with data from spiked samples, as the results differed significantly from those obtained using cancer patient samples. This was also stated more explicitly in the manuscript [Lines 141 – 146 of the revised manuscript]. The differences can be explained by the fact that it is difficult to mimic the fraction of DNA trapped in or adhered to vesicles and/or exosomes. We also expanded the material and methods section to state that the concentration of plasma samples was not determined before ddPCR [Lines 372 – 373 of the revised manuscript].
2. Please improve English
As per reviewers' suggestions, the manuscript has been adjusted by a native English speaking colleague.
Taken together, our manuscript in its revised form has clearly benefited from the reviewer's profound evaluation of the paper. We hope to have convinced you to reconsider our manuscript for publication in the special issue "Liquid Biopsy for Cancer" in your journal Cancers.
Yours sincerely
Laure Sorber (corresponding author)
Prof. Dr. Patrick Pauwels (senior author)

Reviewer 2 Report
In this study by Sorber L. et. al. the authors isolated the cfDNA and cfRNA from peripheral blood (PB) collected in EDTA and Streck tubes. For separating plasma from the blood specimens, the authors examined varying centrifugation conditions (speed, time, temperature, and steps) and identified the conditions that could yield cfDNA and cfRNA from EDTA tubes. Interestingly, the authors noticed Streck tubes could yield only cfDNA but not the cfRNA at all tested centrifugation conditions for isolating plasma from PB . Identifying critical conditions that could yield both cfDNA and cfRNA from single blood specimen will have potential benefit to the emerging applications of liquid biopsy in cancer patients’ management. This article appears to be written well, however, there remain many major concerns the authors should address.
Sample size used in this study is very small; the authors should strengthen their observations by using a larger sample cohort that is statistically significant where ever appropriate.
It is not clear whether the centrifugation conditions that the author identified from this study are relevant to cfDNA isolation methods that they have used or it can be generalized to other extraction methods as well. The authors should discuss and at least repeat with other extraction methods for a small batch of the samples.
Except CPStreck, CPCEN, and CPAdCEN, in other protocols (CPBasic etc) very low speed centrifugation was used. How did the authors ensure the recovery of cell free plasma in those centrifugation protocols?
The figure legends need to be more descriptive. The authors should provide additional information, e.g. were healthy donor samples or cancer patients’ samples used? What is the sample size of the chosen experiment?
Figure1: The authors should clarify whether the same patient sample is collected in both EDTA and Streck tubes and used to determine total yield and DNA integrity.
Figure3: Why did the authors choose different patients for the EDTA vs Streck tube comparison? The sample size is very small for each group; the authors should add more samples. In the case of patient 8, mutant copies abundance is not correlated with allele frequency, how can that be explained?
Figure3 and 5: It is not clear whether the same patients’ samples were used for both figures? If the same samples are used the allelic frequencies observed from cfDNA are not concordant with those from cfRNA. How can this be explained? The total cfRNA copies obtained per ml of blood appears to be several folds lower than the cfDNA copies obtained from the same amount of blood. If this is true what potential applications do the authors foresee for cfRNA? The authors should include this as part of the discussion.
Table2: should be simplified. Numbers mentioned under ‘specifications’ should be detailed in legend.
Author Response
Dear Editor
Dear Reviewer
Please find enclosed the amended version of our manuscript 'Circulating Cell-Free DNA and RNA Analysis as Liquid Biopsy: Optimal Centrifugation Protocol'. We very much appreciate the constructive comments of the reviewer.
We are pleased to submit the improved manuscript within the requested deadline. We have adapted our manuscript according to the reviewer's recommendations, as described by our point-by-point reply below:
1. Remarks concerning sample size
a. Sample size used in this study is very small; the authors should strengthen their observations by using a larger sample cohort that is statistically significant where ever appropriate.
b. Figure3: The sample size is very small for each group; the authors should add more samples.
As the reviewer correctly points out, our sample size is relatively small. However, our statistical analysis of cfDNA and cfRNA concentration and cfDNA integrity demonstrates in strong significant p-values, detailed in Supplementary Table 1. This was further confirmed by the data from the KRAS mutation detection (by digital droplet PCR), despite the low number of patients with detectable KRAS-mutated ctDNA/ctRNA. Due to ethical and practical considerations, it is difficult to request more patients to give 50 mL of blood when our data show strong statistical difference. As the low sample size is an important issue, we have explicitly stated this in the revised manuscript [Lines 186 – 187 of the revised manuscript].
2. It is not clear whether the centrifugation conditions that the author identified from this study are relevant to cfDNA isolation methods that they have used or it can be generalized to other extraction methods as well. The authors should discuss and at least repeat with other extraction methods for a small batch of the samples.
The reviewer has indicated an important issue. In 2016, we performed a comparison study of five different cfDNA isolation kits [Sorber et al. 2017, Journal of Molecular Diagnostics]. One of the main findings of this study was that the Maxwell RSC ccfDNA Isolation Kit (Promega) had a similar isolation efficiency as the QIAamp Circulating Nucleic Acid Kit (Qiagen), which is one of the most commonly used cfDNA isolation kits. This was also reported in another study [Pérez-Barrios et al. 2016, Transl Lung Cancer Res]. Despite the fact that these cfDNA isolation kits differ in cfDNA capture method (column- versus magnetic beads based), they both were shown to favor short cfDNA fragments and their protocols contain a lysis step before cfDNA binding. In this manner, they were both able to capture nucleic acids trapped in or adhered to vesicles and/or exosomes. We also regularly compare these kits to newly developed cfDNA isolation kits. Hence, we are confident that our technique is up to date and that our findings can be generalized. The detection of cfRNA in plasma or adhered to platelets could differ from our findings with the development of novel cfRNA isolation kits (which are currently being evaluated in a Belgian study). We have added a sentence in the discussion that the findings of this study will aid in the standardization of the isolation and analysis of both nucleic acids [Lines 275 – 276 of the revised manuscript].
3. Except CPStreck, CPCEN, and CPAdCEN, in other protocols (CPBasic etc) very low speed centrifugation was used. How did the authors ensure the recovery of cell free plasma in those centrifugation protocols?
In this study, we selected centrifugation protocols (from other research groups) that have already been established to be compatible with cfDNA (and cfRNA) analysis. Adapted versions of some of these protocols were included for practicality (samples at room temperature) or to conform with current recommendations (two centrifugation steps). We have expanded the material and methods section to further clarify the selected centrifugation protocols [Lines 342 – 357 of the revised manuscript]. Despite the use of low centrifugation speed, no difference in plasma volume was noted (data not shown). Hence, the isolated cfDNA and cfRNA could be compared directly between the different centrifugation protocols. As this is an important point, we have included this in the discussion [Lines 228 – 232, 234, and 235 – 236 of the revised manuscript]. As expected, the low speed centrifugation protocols, with the exception of the platelet-generating protocol, were found to contain more long DNA fragments (> 400 bp), indicating genomic DNA contamination. The cfDNA quality was investigated by using the integrity index, as we expected that more gDNA contamination would be present in these centrifugation protocols. We did not expect that the platelet-generating protocol would provide good quality cfDNA, as highlighted in the discussion.
4. The figure legends need to be more descriptive. The authors should provide additional information, e.g. were healthy donor samples or cancer patients’ samples used? What is the sample size of the chosen experiment?
As per reviewers' suggestion, we have adapted all figures and figure legends to provide additional information regarding sample size.
5. Remarks concerning EDTA and Streck tubes.
a. Figure1: The authors should clarify whether the same patient sample is collected in both EDTA and Streck tubes and used to determine total yield and DNA integrity.
b. Figure3: Why did the authors choose different patients for the EDTA vs Streck tube comparison?
The same patient samples were not collected in both EDTA and Streck tubes, as this was not possible due to practical and ethical considerations. Participants only provided 50 mL of blood samples in one type of blood collection tubes. Moreover, the comparison of the performance of EDTA and Streck tubes was not part of the scope of this study. As the aim was to determine the effect of centrifugation protocol on cfDNA and cfRNA yield and quality. In order to further clarify this, we have rewritten this part of "4.1. Study design" [Lines 324 – 325 of the revised manuscript]. Furthermore, the legend of Figure 1 has been altered to explicitly state that in all samples total cfDNA concentration and cfDNA integrity has been investigated [Lines 102 – 105 of the revised manuscript]. We also included a phrase in which we explicitly state that the comparison of EDTA and Streck tubes is not part of the aim of this study [Lines 122 – 126 of the revised manuscript].
6. Figure 3: In the case of patient 8, mutant copies abundance is not correlated with allele frequency, how can that be explained?
The exact mechanism of cfDNA and ctDNA release in the circulating remains unknown. However, increased total cfDNA concentrations have been demonstrated in certain disorders and physiological processes, such as inflammation, pregnancy, myocardial infarction, and autoimmune disorders [Volckmar et al. 2018, Genes Chromosomes Cancer]. In the case of patient 8 (patient 7 in the revised manuscript) a biological phenomenon is most likely responsible for the low allele frequencies. We are confident that these are not the result of gDNA contamination, especially in case of the platelet-generating, and the two-step, high-speed protocols. In these protocols, the cfDNA integrity analysis indicated a rather low percentage of long fragments. We thank the reviewer for bringing up this important issue and have commented upon this phenomenon in the discussion [Lines 223 – 227 of the revised manuscript].
7. Figure3 and 5: It is not clear whether the same patients’ samples were used for both figures? If the same samples are used the allelic frequencies observed from cfDNA are not concordant with those from cfRNA. How can this be explained? The total cfRNA copies obtained per ml of blood appears to be several folds lower than the cfDNA copies obtained from the same amount of blood. If this is true what potential applications do the authors foresee for cfRNA? The authors should include this as part of the discussion.
We have adapted figures 3 and 5 to more explicitly show which patients are represented (patients 1 - 4 in EDTA and patients 5 – 8 in Streck tubes, respectively).
As the reviewer correctly pointed out, there is a marked difference in mutated ctRNA concentration and allele frequency. We have adapted the discussion to comment on the fact that further research is needed to ensure successful cfRNA analysis as it seems prone to contamination with RNA from residual cells and microparticles [Lines 276 – 286 of the revised manuscript]. The true potential of cfRNA analysis is the detection of gene fusions and amplifications. These are difficult to detect in cfDNA, while this is a more convenient matrix for mutation detection. We have added a sentence in which we explicitly state the potential of cfRNA analysis [Lines 289 – 291 of the revised manuscript].
8. Table2: should be simplified. Numbers mentioned under ‘specifications’ should be detailed in legend.
We would like to thank the author for pointing out our error with regards to the numbering in 'specifications'. We have corrected this and added details to the table legend [Line 361 of the revised manuscript].
Taken together, our manuscript in its revised form has clearly benefited from the reviewer's profound evaluation of the paper. We hope to have convinced you to reconsider our manuscript for publication in the special issue "Liquid Biopsy for Cancer" in your journal Cancers.
Yours sincerely
Laure Sorber (corresponding author)
Prof. Dr. Patrick Pauwels (senior author)
